# The evolution of bone-eating worm diversity in the Upper Cretaceous Chalk Group of the United Kingdom

Sarah Jamison-Todd[1,2]*, James D. Witts[2], Marc E. H. Jones[2], Deborah Tangunan[1], Kim Chandler[2], Paul Bown[1], Richard J. Twitchett[2]

**1** University College London, London, United Kingdom, **2** Natural History Museum, London, United Kingdom

* sarah.jamison-todd@nhm.ac.uk

## Abstract

The bone-eating worm *Osedax* is today a member of the highly adapted invertebrate assemblages associated with whale carcasses on the ocean floor. The worm has also been found in a variety of other vertebrates in marine environments. *Osedax* borings are represented by the trace fossil *Osspecus*, which has been identified in fossil whales and marine reptiles, with the earliest occurrence in the Albian. In studies of present-day whale bones it has been found that individual species of *Osedax* create distinct boring morphologies. The diversity of boring geometries therefore provides a proxy for species diversity that can be applied to the fossil record to better understand the diversity, ecology, and evolution of extinct *Osedax* species. We examined marine reptile fossils from the Upper Cretaceous Chalk Group of the United Kingdom, and found five previously undocumented boring morphologies. These results, coupled with a re-examination of previous records of *Osspecus,* led to the naming of seven new ichnospecies. Using nannofossil biostratigraphy from the chalk, we constrained the ages of these occurrences and found a high species diversity in the early Late Cretaceous, indicating either a rapid diversification or an earlier origin for *Osedax* than previously estimated. Furthermore, we recognise five Cretaceous ichnospecies that are also found in the Cenozoic, three of which are also found in present-day whale bones.

## Introduction

The siboglinid annelid worm *Osedax* is distinguished by its unique adaptations to digest bone in marine environments and its status as a member of the often highly derived and unique invertebrate communities surrounding whale-falls [1], though it is also found on a variety of other bones [2]. Members of this genus lack a mouth and gut and digest lipids and/or collagens with the aid of acquired bacterial symbionts [3–5]. There are 34 described living species of *Osedax* [2,6] (World Register of Marine Species [WoRMS]; https://marinespecies.org), but this number is likely to increase. Species are known not only from natural whale-falls but also from whale carcasses experimentally deployed on the ocean floor and bones and tissues belonging to a number of other vertebrates [2]. Additional species that have been collected

**Data availability statement:** The CT scans and images of the location in the bone of the holotypes are available on MorphoSource under the project title 'Wormy Chalk': (https://www.morphosource.org/projects/000651498?locale=en). Scans can be downloaded from this link.

**Funding:** This study was supported in part by the NERC-funded ChaSE (Chalk Sea Ecosystems) project (NE/X015300/1, NE/X015386/1). Natural Environment and Research Council website: https://www.ukri.org/councils/nerc/. The grant was awarded to RJT and PB. The funders had no role in study design, data collection and analysis, decision to publish, or preparation of the manuscript.

**Competing interests:** The authors have declared that no competing interests exist.

remain unpublished or undescribed [2]; some pending descriptions are in preparation. Known species are concentrated in regions which have experienced targeted research efforts into discovering whale-falls and describing the associated fauna, but *Osedax* has been found in all major ocean basins from the Arctic [7] to the Antarctic [8] (Global Biodiversity Information Facility [GBIF]; https://www.gbif.org). The broad range of water depths inhabited by *Osedax* (from ~ 20 m [8] to ~ 4200 m [9]) shows additional adaptability of these organisms to a wide range of environments in today's oceans.

Characteristic anatomical features of *Osedax* worms include a trunk sitting within a borehole aperture and branching ovisac structures expanding beneath the bone surface [3]. This anatomy is associated with a unique boring morphology that has been described both in modern and fossil bones, and which is assigned to the ichnogenus *Osspecus* [10] (Fig 1). One ichnospecies, *Osspecus tuscia*, has previously been referred to this genus [10]. No other organisms are known to make borings in bone with this distinct morphology. CT-scanning may be the only reliable non-destructive method for identification of *Osedax* borings, as the morphology of the internal chamber is not clear when only the aperture is visible. Even where the bone surface and aperture have been lost, due to subsequent erosion or bioerosion or due to poor preservation, CT-scanning may assist in inferring an *Osedax* tracemaker for less well-preserved, exposed, or partial chambers. Proximity of and similarity to complete chambers nearby in the bone may also be used to infer an *Osedax* tracemaker for these less well-preserved chambers. Modern borings described through CT scanning are generally chambers ~ 3–10 mm in diameter with a single entrance hole of ~ 1 mm diameter [11,12]. Fossil examples described in comparison also fit these parameters. Examples confirmed via CT-scanning include instances in whale bones from the Pliocene of Italy [10] and whale and fish bones from the Oligocene of Washington, USA [13,14]. *Osspecus* has been identified in plesiosaur and sea turtle bones from the Albian of the United Kingdom [15], from a plesiosaur tooth from the Cenomanian of the United Kingdom, from plesiosaur bones from the Campanian of North America, and mosasaur bones from the Maastrichtian of Belgium [16].

The most recent origin and diversification estimates for *Osedax* use the earliest fossil record of *Osspecus* in the Albian [15] and the earliest fossil evidence for the sister group of vestimentiferan tube worms in the Campanian [17] and place the origination of this clade in the late Early Cretaceous, with diversification progressing through and after the Late Cretaceous [18]. These estimates have severe limitations, however, and serve as the youngest time frame

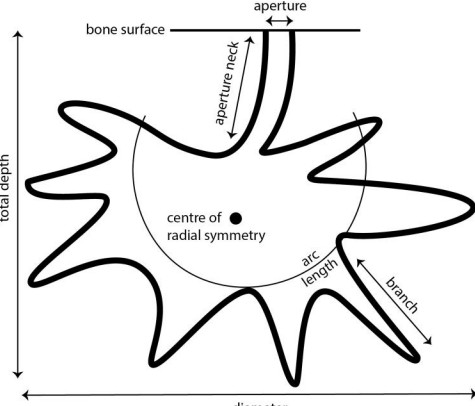

**Fig 1. Diagram of idealised *Osedax* boring showing the broad morphological features used to define morphotypes.**

estimates for origin and diversification, given that these dates coincide with the earliest fossil evidence, which is unlikely to be the earliest actual occurrence of *Osedax* or its osteophagous ancestors.

Modern descriptions of *Osspecus* not only provide benchmarks for the identification of fossil *Osedax* borings but have also shown that there is a correlation between different boring morphotypes and the species of *Osedax* that creates them [11,12]. Borings made by the same species show some degree of variation in size and depth, as the borehole structure is affected by differences in bone density, porosity, and postmortem taphonomy of the bone. Nevertheless, where distinct morphologies are created by a particular extant species, these morphologies are generally consistent between bone elements [11,12]. This relationship between species and boring geometry is also supported by fossil instances that show one distinct boring morphology throughout multiple elements of the same skeleton [16]. Using boring morphotypes as a proxy, it has been shown that multiple species of *Osedax* were likely present in the Late Cretaceous, and on both sides of the North Atlantic Ocean basin [16]. These records suggest that taxonomic diversity may have already been relatively high in the Late Cretaceous, and geographic distribution broader than previously understood [16].

Here we analyse *Osspecus* ichnospecies in specimens of plesiosaur, mosasaur, and ichthyosaur bones to better constrain the relative species diversity of *Osedax* in the Late Cretaceous. Using nannofossil dating of the sedimentary rock surrounding the specimens, all of which are sourced from the Chalk Group of the United Kingdom [19], we are able to infer an accurate geological age for each specimen and therefore better understand the temporal distribution of *Osspecus* through the Late Cretaceous than has been previously possible.

## Materials and methods

### Materials

Specimens used for this study were sourced from the marine reptile collection of the Natural History Museum, London (NHMUK). Approximately 130 mosasaurs, plesiosaurs, and ichthyosaurs from the UK Chalk Group were examined for surface bioerosion bearing a resemblance to the apertures or weathered chambers of borings known to be made by extant *Osedax*. The Chalk Group was targeted because specimens could be accurately dated using calcareous nannofossil biostratigraphy of the adhering chalk matrix. Of the 130 examined, six marine reptile specimens, comprising isolated bones and teeth with associated chalk matrix, are presented here (Table 1). *Osspecus* has already been described in another marine reptile from the Chalk, NHMUK PV R 35103 [16], and these six additional specimens showed the best evidence for additional examples of *Osspecus*.

**Table 1. The marine reptile specimens with *Osspecus* newly described in this study, including all available information on provenance.**

| Registration number | Element | Location (UK) | Taxon |
|---|---|---|---|
| NHMUK PV R 1233 | dorsal vertebra | unrecorded | Mosasauria indet. (mosasaur) |
| NHMUK PV R 3355 | partial right maxilla bearing three teeth | Cuxton, Kent | Russellosaurinae indet. (mosasaur) |
| NHMUK PV R 1215 | tooth | Kent | Pliosauridae indet. (plesiosaur) |
| NHMUK PV R 35103 | tooth | Rochester, Kent | Pliosauridae indet. (plesiosaur) |
| NHMUK PV R 4205 | propodial (unfused) | unrecorded | Plesiosauria indet. (plesiosaur) |
| NHMUK PV R 1265 | five teeth | Lewes, East Sussex | Pliosauridae indet. (plesiosaur) |
| NHMUK PV OR 32812 | partial jaw | Round Down tunnel, Kent | *Pervushovisaurus campylodon* (ichthyosaur) |

## CT-scanning

Bones were selected for scanning based on surface erosion or bioerosion resembling *Osspecus* (Fig 2). All specimens were scanned using the Nikon Metrology HMX ST 225 micro-CT scanner, and scans were reconstructed using the software Avizo (FEI Visualization Science Group; https://www.thermofisher.com). Borings visible on the scans were sorted based on distinct geometries into morphotype groups and the range of diameter and depth was measured for each morphotype. The best example of each morphotype was segmented and reconstructed three-dimensionally and individual measurements were taken of these type borings (Fig 3, Table 2). Boring types were classified based on gross morphology, using features that are distinct between morphotypes. These features include diameter range, depth range, aperture diameter relative to chamber size, the relative depth at which a chamber sits within the bone, centre of primary symmetry, presence of secondary radial symmetry, branch length relative to chamber size, branch shape, and overall chamber shape (Fig 1). The depth is defined as the total depth of the boring, from aperture to the base of the chamber. Surface penetration is defined as 'shallow' if the chamber sits close enough to the surface to be nearly touching it, 'mid' if it sits under the surface but has an aperture neck not longer than the main chamber depth, and 'deep' if the aperture neck is longer than the chamber depth. Branch length is defined as 'mid' if the chamber branches approximately halfway from its centre to the outer reach of the branches, as 'long' if the branching point is interior to this halfway point, and 'short' if exterior to this halfway point. Arc length is an additional parameter that may be useful, although it can be variable, and is defined as the approximate arc in degrees for the sector curve of a cross-sectional boring. The term 'inward' refers to the direction penetrating into the bone, and 'outward' as moving away from the centre of the main boring chamber. Relative size of the borings to each other may also be a useful parameter, though for some ichnospecies size of the borings may also be variable, and there may

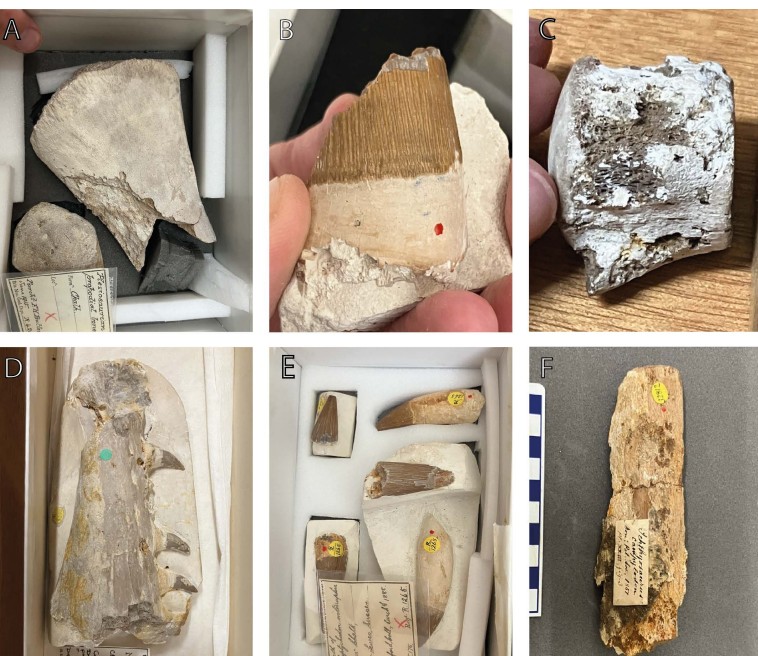

**Fig 2. Surface view of specimens with *Osspecus* bioerosion.** A) NHMUK PV R 4205. B) NHMUK PV R 1215. C) NHMUK PV R 1233. D) NHMUK PV R 3355. E) NHMUK PV R 1265. F) NHMUK PV OR 32812.

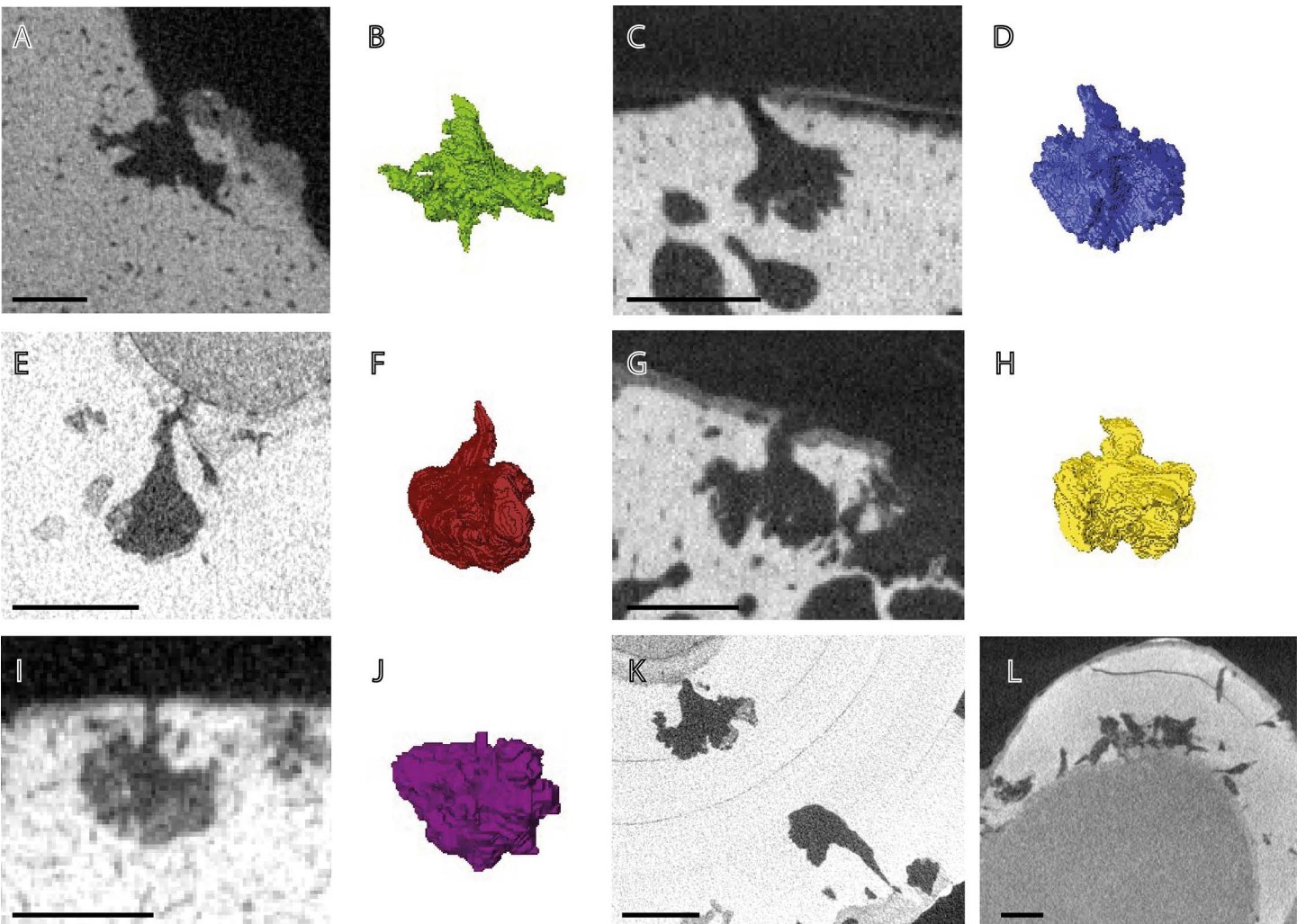

**Fig 3. The 2D cross-sections and 3D reconstructions of the type borings of each of the five morphotypes present in the specimens addressed in this study.**
A) Cross-section of an *O. eunicefootia* boring in NHMUK PV OR 32812. B) Three-dimensional reconstruction of the boring in panel A. C) Cross-section of an *O. tuscia* boring NHMUK PV R 1233. D) Three-dimensional reconstruction of the boring in panel C. E) Cross-section of an *O. morus* boring in NHMUK PV R 1215. F) Three-dimensional reconstruction of the boring in panel E. G) Cross-section of an *O. campanicum* boring in NHMUK PV R 1233. H) Three-dimensional reconstruction of the type boring in panel G. I) Cross-section of an *O. arboreum* boring in NHMUK PV R 4205. J) Three-dimensional reconstruction of the boring in panel I. K) Different morphologies of *O. morus* borings in NHMUK PV R 1215. L) Tubular bioerosion in NHMUK PV R 1265 that may be exploratory and incomplete *Osedax* borings or another form of bioerosion. Scale bars are 2 mm in all panels.

be overlap in size ranges between ichnospecies. Where proportions of individual ichnospecies are described in relative terms, refer to Table 2 for the exact dimensions of the holotype borings.

## Nannofossil biostratigraphy

Chalk is a biogenic sediment predominantly composed of calcareous nannofossils which are the micron-scale remains of coccolithophorid phytoplanktonic algae. Their abundance, diversity and evolutionary turnover through the Cenomanian–Maastrichtian chalk succession of NW Europe makes them ideal biostratigraphic fossils facilitating dating to geological substage level, with 20 recognized biozones in the Late Cretaceous [20]. Many of the marine reptile specimens in older collections such as those at NHMUK have limited associated stratigraphic information, and this method provides refined age data for specimens that are otherwise

**Table 2. The *Osspecus* ichnospecies presented in this study and the characteristic features that were used to define them.**

| ichnospecies | specimens with ichnospecies | diameter range (mm) | depth range (mm) | type diameter (mm) | type depth (mm) | type aperture (mm) | surface penetration | centre of radial symmetry | secondary symmetry | branch length | branch shape | chamber shape | holotype reference | holotype accession number | in vertebrate specimen |
|---|---|---|---|---|---|---|---|---|---|---|---|---|---|---|---|
| *Osspecus tuscia* | NHMUK PV R 1233; NHMUK PV R 1265; NHMUK PV OR 32812 | 0.7–3.7 | 0.6–2.1 | between 1.7 and 3.7 | unknown | between 0.6 and 1.2 | mid | aperture neck base | no | mid | lobate with pointy ends | sector | Higgs et al. (2012) | n/a | IGF 1134T |
| *Osspecus eunicefootia* | NHMUK PV OR 32812 | 1.8–2.7 | 1.2–1.9 | 3.5 | 2.7 | 0.7 | shallow | top of chamber | no | long | wavy and thin with pointed ends | hemispherical | this study | NHMUK PX TF 309 | NHMUK PV OR 32812 |
| *Osspecus morsus* | NHMUK PV R 1215; NHMUK PV R 3355; NHMUK PV R 35103 | 1.3–2.4 | 2.3–3.7 | 1.6 | 2.5 | 0.2 | mid to deep | aperture neck base | possibly | short | lobate with indistinct ends | sector | this study | NHMUK PX TF 310 | NHMUK PV R 1215 |
| *Osspecus campanicum* | NHMUK PV R 1233 | 0.5–2.4 | 0.5–2.1 | 2.4 | 2.1 | 0.5 | mid | upper chamber | possibly | mid | rounded with lobate ends | irregular rounded | this study | NHMUK PX TF 311 | NHMUK PV R 1233 |
| *Osspecus arboreum* | NHMUK PV R 4205; NHMUK PV R 3355 | 1.1–2.3 | 1.4–2.6 | 2 | 2.2 | 0.2 | mid | upper chamber | no | very short | rounded | round | this study | NHMUK PX TF 312 | NHMUK PV R 4205 |
| *Osspecus automedon* | NHMUK PV R 35103 | 0.5–3 | 1–5 | 3.1 | 4 | 0.6 | mid to deep | aperture neck base | possibly | very long | spoke with bulb at end | rounded sector | Jamison-Todd et al. (2024) | NHMUK PX TF 313 | NHMUK PV R 35103 |
| *Osspecus frumentum* | NHMUK PV R 35103 | single boring | single boring | 1 | 1.5 | 0.1 | mid | central axis of chamber | no | long | threadlike | columnar with offshoots | Jamison-Todd et al. (2024) | NHMUK PX TF 314 | NHMUK PV R 35103 |
| *Osspecus panatlanticum* | FMNH PR 187; IRSNB 369; IRSNB R 370 | 1–3 | 0.5–1.5 | 0.9 | 1.3 | 0.2 | shallow | centre of chamber | yes | short | narrow with rounded ends | hemispherical | Jamison-Todd et al. (2024) | IRSNB 7767 | IRSNB R 370 |

labelled with very broad stratigraphic terms, e.g., 'Upper Cretaceous', 'Chalk', or historical subdivisions 'Lower', 'Middle', or 'Upper' Chalk which have been superseded by more detailed lithostratigraphic and biostratigraphic schemes [19,20].

Chalk samples for biostratigraphic analysis were taken from the fossil specimens that showed *Osedax* bioerosion confirmed by CT scanning, including the previously described NHMUK PV R 35103 [16]. Matrix was sampled away from any bone or other fossil material, to avoid damaging the specimens. A small surface area of the chalk matrix surrounding the bone was scraped off and discarded using a scalpel or wooden toothpick. The uncontaminated chalk underneath was then gently scraped until a sample of between 0.02–0.1 g was collected. Fresh scalpels or toothpicks were used between specimens to avoid contamination.

A nannofossil smear slide for light microscope analysis was prepared using the standard method outlined in Bown & Young (1998) [21]. A small drop of distilled water was placed on a glass coverslip. The powdered sediment was mixed with the water using a toothpick and spread evenly across the coverslip. The coverslip was dried on a hotplate and then fixed to a glass microscope slide using Norland Optical Adhesive. The adhesive was then cured in a UV light box for approximately 15 minutes.

The slides were analysed using a polarising light microscope at 1000x magnification. The nannofossils were logged across 100 fields of view (FOVs), followed by a further 100 FOVs, where any additional species were noted. The presence or absence of biostratigraphic index marker taxa were used to determine the nannofossil zone or subzone. The Upper Cretaceous (UC) nannofossil zonation scheme of Burnett et al. (1998) [20] was applied and taxonomy generally followed Nannotax (http://ina.tmsoc.org/Nannotax3) [22]. Age calibrations of the zonal/subzonal bioevents follow Gradstein et al. (2020) [23]. Where location metadata are available, we also assign specimens to geological formation [19].

## Results

The six bones had varying degrees of bioerosion visible on the surface (Fig 2) and within (Fig 3), and show a variety of borings. The borings are in a range of marine reptile bones: three plesiosaurs, two mosasaurs, and one ichthyosaur, which provides the first described instance of *Osspecus* in an ichthyosaur. Along with previously recognised boring geometries from Jamison-Todd et al. (2024) [16], we define seven new ichnospecies of *Osspecus*. We also recognize additional examples of *O. tuscia*, previously described only in Pliocene whalebone [10], in this new material. Five of the now eight ichnospecies of *Osspecus* are present in the six marine reptile specimens newly presented here.

### Systematic ichnotaxonomy

*Osspecus* igen. Higgs et al. 2012 [10]

Holotype: Boring number 1 (Figs 3(a) and 4(a–c)) in a fossil cetacean radius stored in the Museo di Storia Naturale, Sezione di Geologia e Paleontologia, Florence, Italy (IGF 1134T).

**Diagnosis.** Single entry borings found in bone substrates. Individual borings consist of a circular to sub-circular aperture, lacking any rim or platform. This aperture extends into the bone as a uniformly thick canal, generally perpendicular to the bone surface, with a globular or irregularly shaped chamber at the base of the canal. These chambers may or may not have thin exploratory tunnels emanating from them.

*Osspecus tuscia* isp. Higgs et al. 2012 [10]

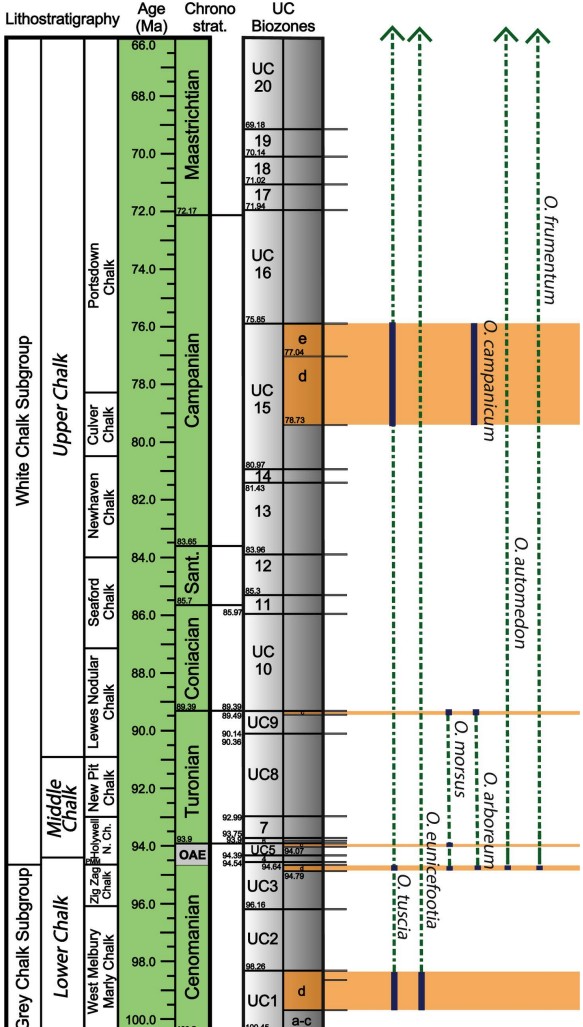

**Fig 4. Litho- and biostratigraphic scheme for the UK Chalk Group, showing the succession of calcareous nan-
nofossil zones, and current lithostratigraphic divisions.** Constrained marine reptile specimen dates are marked
by orange bars. Ichnospecies occurrences in the Upper Cretaceous are indicated by navy bars, and inferred ranges
by dashed green lines. Ichnospecies that are found in the Cenozoic have an inferred range above the Cretaceous,
indicated by arrows. Nannofossil zonation scheme is modified from Burnett (1998) [20] and lithostratigraphy is mod-
ified from Rawson et al. (2001) [26]. Note that *O. panatlanticum* is not found in the UK Chalk. Numerical ages from
GTS2020 [27] and TimescaleCreator8 (timescalecreator.org).

Holotype: same as genus

   **Diagnosis** [10]**.** Boring with sub-millimetre-sized apertures. The base of the apertural
canal tapers into a chamber that is partially flattened in the vertical plane. Short globular
exploratory lobes extend from the main body of the chamber.

   **Emended diagnosis.** Boring with chamber diameters of 0.7–2.0 mm and total depth of
0.6–2.1 mm. The wide aperture leads to a tapering aperture neck that is wider at the base near
the top of the chamber. The chamber sits at mid-depth below the surface of the bone. Radial
symmetry is centred at the base of the aperture neck, and no secondary symmetry is evident.
Branches are of mid-length relative to the chamber and maximum arc length is 180 degrees
or less, so that the branches generally point outwards or inwards but not up towards the bone

surface. Branch shape is lobate and also tapering, with branches having pointed ends and widening into the chamber.

**Remarks.** The measured borings from Higgs et al. (2012) [10] are larger, but they overlap in size ranges with those measured for this study. The lesser arc length and tapering nature of the branches and aperture neck gives the borings the appearance of being slightly vertically stretched, but some of the borings in Higgs et al, though sharing the other features that define this morphotype, are vertically flattened. The original description as it stands therefore encompasses some variant morphologies. Here, the more vertically stretched endmember with a shorter arc length is predominant. This ichnospecies is present in three marine reptile specimens. Figured here in Fig 3, panels C and D.

*Osspecus eunicefootia* isp. nov.

**Etymology.** In honour of Eunice Newton Foote (1819-1888), the first person to suggest on experimental grounds that an increase in atmospheric $CO_2$ would in turn increase the temperature of the Earth [24,25]. She was also an inventor and campaigned for women's rights [25].

**Holotype.** NHMUK PX TF 309
Fig 3 this study, panels A and B, boring in NHMUK PV OR 32812

**Additional examples.** Jamison-Todd et al. 2024 [16], Fig 2, panels G and H. Higgs et al. (2014) [12], Fig 3.

**Diagnosis.** Boring with chamber diameters of 1.8–2.7 mm and total depths of 1.2–1.9 mm. Apertures are wide relative to other boring types. Borings are shallow, the main chamber sitting just below the surface of the bone, with a short aperture neck. The centre of radial symmetry sits at the top of the chamber. Branches are wavy, thin, and long relative to the chamber, though they can be of irregular length. Overall chamber shape is approximately hemispherical, with an arc length generally close to 180 degrees.

**Remarks.** The chambers are large relative to most other *Osspecus* ichnospecies. Due to the shallow depth of the hemispherical chambers they are prone to collapse and weathering, with the interior of the chamber exposed and the outer layer of bone covering the boring unpreserved. These borings are found in one marine reptile specimen in this study, and were previously described in Jamison-Todd et al. (2024) in a plesiosaur from North America, which represents a particularly large example of this boring type [16]. This morphotype was therefore present on both sides of the Atlantic Ocean basin in the Cretaceous. The living species *Osedax antarcticus* also creates borings that can be referred to this new ichnospecies [12].

*Osspecus morsus* isp. nov.

**Etymology.** After Latin for 'bite', in double reference to the eating of the bones and the fact this boring type is usually found in teeth.

**Holotype.** NHMUK PX TF 310
Fig 3 this study, panels E and F. Boring in NHMUK PV R 1215.

**Additional examples.** Jamison-Todd et al. (2024) [16], Fig 2, panels C and D.

**Diagnosis.** Borings with chamber diameters ranging from 1.3–2.4 mm, and total depths from 2.3–3.7 mm. Aperture necks can be variable in length. Chambers that are set deeper within the bone are connected to long, curved aperture necks leading to proportionally small chambers. The centre of radial symmetry of individual chambers is set at the base of the aperture neck, and there are commonly clusters of branches within these chambers, with individual radial

symmetry. Branches are small and lobate and can create a ridged appearance around the edge of the chamber. Arc length is also variable, with some chambers branching widely over 180 degrees and others branching in a narrow minor arc of less than 90 degrees, giving the chambers a triangular cross-sectional shape that sometimes spreads into a broader arc.

**Remarks.** This morphotype is the most variable in appearance and broad morphology. Some of the variability in this boring morphotype is figured here in Fig 3, panel K. This type of boring is commonly but not always found in the dentine at the base of teeth, occasionally puncturing the enamelled areas. The boring depth and the curvature of the aperture necks may be related to density differences within the dentine, but the histology of dentine is poorly documented in reptiles. It is also possible that the morphological variability of this boring type may represent the borings of multiple species. It is also not clear whether the clusters of chambers are made by the same animal as a result of secondary symmetry in the morphology of the worm, or if multiple animals are exploiting the same point of entry into the bone. The commonalities between certain features of these borings, however, lead to their classification into one type. Borings previously described in Jamison-Todd et al. (2024) as 'type 2' also belong to this ichnospecies and co-occur in one tooth with two other ichnospecies [16]. These borings are present in two of the specimens presented here.

*Osspecus campanicum* isp. nov.

**Etymology.** This is the only ichnospecies found exclusively in the Campanian Stage of the Late Cretaceous.

**Holotype.** NHMUK PX TF 311
Fig 3 this study, panels G and H. Boring in NHMUK PV R 1233.

**Diagnosis.** Boring with chamber diameters ranging from 0.5–1.4 mm, and total depth of 0.5–2.0 mm. Apertures are wide and the aperture necks leading into the chambers are columnar. The chambers sit at mid-depth in the bone. The centre of symmetry is not always distinct, though it generally sits towards the upper middle of the chamber. Symmetry is not regularly radial, with the branches pointing mostly inwards, and the presence of secondary symmetry in some clusters of branches is possible. Chambers are therefore irregularly shaped, and the branches are not of equal size and length. Branch shape is lobate and rounded. Arc length is approximately 180 degrees.

**Remarks.** The irregular rounded branches give this boring type a squat cartoonish appearance. Smaller borings may make clusters that create larger pits when they are collapsed or weathered. These borings share some features with *O. tuscia*, and are of a similar scale, the main differences in *O. campanicum* being the wider aperture and the rounded shape of the lobate branches; neither aperture nor branches have a tapering appearance as in *O. tuscia* [10]. This morphotype is found here in one specimen.

*Osspecus arboreum* isp. nov.

**Etymology.** For its resemblance to a typical tree shape. In clusters the borings resemble tiny forests in cross section.

**Holotype.** NHMUK PX TF 312
Fig 3 this study, panels I and J, boring in NHMUK PV R 4205.

**Diagnosis.** Boring with chamber diameters ranging from 1.1–2.3 mm and total depths 1.4–2.6 mm. Aperture is fine and straight, with the threadlike aperture neck leading to a chamber that sits at mid-depth beneath the surface of the bone. Chambers have a wide arc and

a globular, regular appearance, with symmetry radiating from the upper chamber. Branches are very short and rounded, merging with and almost indistinct from the main chamber body.

**Remarks.** This form is unusually globular and regular in shape and is more easily distinguished from other *Osspecus* ichnospecies due to this regularity and the very fine threadlike aperture. Complete borings also tend to preserve due to this geometry. This ichnospecies is found in two marine reptile specimens in this study.

*Osspecus automedon* isp. nov.

**Etymology.** After Achilles's charioteer, due to the thin branches with rounded tips merging at the ends, causing the chambers to have the appearance of a segment of a spoked wheel.

**Holotype.** NHMUK PX TF 313
Fig 2, panels A and B, in Jamison-Todd et al. (2024) [16]. Boring in NHMUK PV R 35103.

**Additional examples.** Kiel et al. (2012) [14], Fig 3.

**Diagnosis.** Boring with chamber diameters ranging from 0.5–3 mm and depths from 1–5 mm. Aperture neck length is variable, and therefore depth is also variable. Branches are bar-like and remain somewhat separated until they widen into bulbs and merge at the ends. They are very long relative to the chamber diameter, making up almost all of the chamber. The centre of primary radial symmetry sits at the base of the aperture neck. Arc length can be highly variable; cross sections look like sections of a circle.

**Remarks.** This ichnospecies was previously recorded in the Chalk Group by Jamison-Todd et al. (2024), alongside two other ichnospecies in one tooth [16]. Borings referred to this ichnospecies were also found in an Oligocene whale tooth by Kiel et al. (2012) [14]. No further examples of this ichnospecies were discovered in the marine reptile specimens presented here. While the morphology is distinct, the length of the aperture neck and the arc length of the total chambers appears to be variable. There may be secondary radial symmetry in the branch ends, causing them to broaden at their tips, but it is not clearly visible.

*Osspecus frumentum* isp. nov.

**Etymology.** After the Latin for 'corn', due to its unique symmetry causing this morphotype to resemble a corncob.

**Holotype.** NHMUK PX TF 314
Fig 2, panels E and F in Jamison-Todd et al. (2024) [16]. Boring in NHMUK PV R 35103.

**Additional examples.** Higgs et al. (2014) [12], Fig 2, panels M–R.

**Diagnosis.** Boring where the chamber consists of a central column connected to a tapering aperture neck that is difficult to distinguish from the chamber. Thin filamentous branches extend laterally from the central column of the chamber. Symmetry is linear, rather than radial, in the vertical section, along the main column of the chamber. In horizontal cross section symmetry would be radial around the columnar chamber.

**Remarks.** One chamber was measured at 1 mm diameter and 1.5 mm total depth [16]. This ichnospecies is very easily distinguished from other boring types due to its unique symmetry and resulting distinctive morphology. It occurs in a tooth from the Chalk Group alongside two other ichnospecies [16]. A modern example of this ichnospecies created by *Osedax ryderi* is described by Higgs et al. (2014) [12]. No additional examples are recorded in the present study.

*Osspecus panatlanticum* isp. nov.

**Etymology.** This ichnospecies is the most geographically dispersed in the Cretaceous, as it has been found in specimens from Belgium and the Gulf Coastal Plain of North America, and is one of the two ichnospecies occurring on both sides of the Atlantic during this time.

**Holotype.** IRSNB 7767
Fig 2, panels I and J in Jamison-Todd et al. (2024) [16]. Boring in IRSNB R 370.

**Additional examples.** Higgs et al. (2014) [12], Fig 6
Higgs et al. (2010) [11], Fig 1

**Diagnosis.** Boring where the aperture neck is short and thin when preserved. Chamber depth is shallow and the chambers roughly hemispherical, with an arc length of approximately 180 degrees. Branches extending beyond the main hemispherical chamber are very thin but rounded at the ends. Primary radial symmetry sits in the centre of the chamber, and there is secondary radial symmetry at the ends of the branches, which form clusters and are irregular in length.

**Remarks.** This ichnospecies was called 'type 5' in Jamison-Todd et al. (2024), and was found in two mosasaurs from Belgium, and a plesiosaur from North America [16]. Borings measured in Jamison-Todd et al. (2024) are 1–3 mm in diameter and 0.5–1.5 mm in depth [16], reflecting the range of smaller-scale examples and larger pits that may be created by clusters of weathered borings combining into a larger chamber. It was therefore present on both sides of the Atlantic Ocean basin in the Cretaceous, but has not been recorded in the Chalk Group of the UK. This form is also present in modern whale bones, made by an indeterminate *Osedax* species shown in Higgs et al. (2014) [12], and also the large chambers consisting of overlapping borings created by *Osedax mucofloris* as shown in Higgs et al. (2010) [11]. This ichnospecies is therefore possibly made by two modern species of *Osedax*, though in the former instance the species is unknown, and this could be a case of convergence in boring morphology if this unknown species is not also *Osedax mucofloris*.

## Nannofossil biostratigraphy

Calcareous nannofossils are common to abundant in all the samples and exhibit moderate to good preservation. As is typical for this stratigraphic interval, the zonal/subzonal index species

Table 3. Nannofossil and zonation data for the marine reptile specimens.

| chalk sample number | marine reptile specimen number | age | nannofossil zonation | relevant nannofossil taxa | geological formation |
|---|---|---|---|---|---|
| NHMUK PM NF 6094 | NHMUK PV R 1233 | Upper Campanian | UC15d–eBP | *Prediscosphaera stoveri* (present), *Reinhardtites anthophorus* (present) | Portsdown Chalk |
| NHMUK PM NF 6093 | NHMUK PV R 3355 | Turonian/Coniacian boundary | UC9c | *Broinsonian parca expansa* (present), *Micula staurophora* (absent) | Lewes Nodular Chalk |
| NHMUK PM NF 6120 | NHMUK PV R 1215 | uppermost Cenomanian | UC5c | *Helenea chiastia* (present), *Eprolithus octopetalus* (present) | Holywell Nodular Chalk |
| NHMUK PM NF 6118 | NHMUK PV R 35103 | Upper Cenomanian | UC3d | *Lithraphidites acutus* (present), *Corollithion kennedyi* (present), *Gartnerago nanum* (absent) | Zig Zag Chalk |
| NHMUK PM NF 6119 | NHMUK PV R 4205 | Upper Cenomanian | UC3d | *Lithraphidites acutus* (present), *Corollithion kennedyi* (present), *Gartnerago nanum* (absent) | Zig Zag Chalk |
| NHMUK PM NF 6121 | NHMUK PV R 1265 | Upper Cenomanian | UC3d | *Lithraphidites acutus* (present), *Corollithion kennedyi* (present), *Gartnerago nanum* (absent) | Zig Zag Chalk |
| NHMUK PM NF 6124 | NHMUK PV OR 32812 | Lower Cenomanian | UC1d | *Lithraphidites pseudoquadratus* (present), *Gartnerago segmentatum* (absent), *Lithraphidites acutus* (absent) | West Melbury Marly Chalk |

are rare but, in all cases, we are able to identify a subzone or subzonal range (Table 3, Fig 4). Based on these results, the *Osspecus* trace fossils occur widely within the depositional range of the UK Chalk Group (Fig 4, Table 3). Ichnospecies are not distributed evenly in either space or time, with some ichnospecies dominant in individual specimens and others commonly co-occurring with each other in the same bones (Table 1, Supplementary Data).

## Distribution of ichnospecies

NHMUK PV R 1233: a small vertebra from an indeterminate mosasaur. It appears too incomplete to refer to a specific taxon. Borings are visible at the surface on the sides of the vertebra and can be either intact or partially collapsed (Fig 2). The surfaces that would have connected to other vertebrae are not bioeroded, suggesting that the spinal column may have been in situ or partially still intact when colonized by *Osedax*. The borings are mainly of *O. campanicum*, individual borings combining in some cases to create larger collapsed chambers. There is at least one instance of an *O. tuscia* boring. This specimen was dated to the Upper Campanian.

NHMUK PVR 3355: a partial maxilla bearing three teeth. This specimen was referred to *Mosasaurus gracilis* by Arthur Smith Woodward [28]. However, it is best regarded as an indeterminate russellosaurine mosasaur [28,29]. The borings are visible at the surface as pinholes in the root of the tooth and on the surface of the jaw (Fig 2). Some borehole apertures are covered by sediment. Many borings of *O. arboreum* are present in the jaw section, with additional borings of *O. morsus*. This specimen was dated to the Turonian/Coniacian boundary.

NHMUK PV R 1215: a partial tooth lacking the crown tip from a pliosaurid plesiosaur. The specimen was registered as *Polyptychodon continuus* but this taxon is problematic [30]. The specimen is also likely too incomplete to refer to any species. Therefore, it is best considered Pliosauridae indet. The apertures of uncollapsed borings are visible in the root of the tooth, and on scans are shown to branch both from the inner surface and outer surface of the tooth. Borings stemming from the interior of the tooth are presumed to have been made after the less hard tooth interior had rotted away to some degree. Many of these boring apertures are likely covered in sediment at surface view. The tooth contains borings of *O. morsus*. This specimen was dated to the uppermost Cenomanian.

NHMUK PV R 35103: a tooth from an indeterminate pliosaur. This specimen and its associated bioerosion are described in Jamison-Todd et al. (2024) [16]. The bioerosion is not newly presented here, but is classified here into three ichnospecies, with many examples of *O. morsus*, and one example each of *O. automedon* and *O. frumentum* in this specimen. The specimen is newly dated here to the Upper Cenomanian.

NHMUK PV R 4205: A propodial from an indeterminate plesiosaur. The individual might represent a juvenile as the limb bone is in two pieces and not fused. *Osspecus* is visible both in the limb ends and the more compact bone of the limb. The complete borings are concentrated along an erosional front where one edge of intact cortical bone meets an area of the limb where the cortical bone is completely corroded by collapsed *Osedax* borings (Fig 2). These consist of *O. arboreum*. This specimen was dated to the Upper Cenomanian.

NHMUK PVR 1265: five teeth, two of which include the crown tip. These specimens were registered as *Polyptychodon interruptus* but this taxon is problematic [30] and the specimen is best considered Pliosauridae indet. The teeth sit on small blocks of chalk and show varying degrees of bioerosion. Borings are externally visible as tiny holes in the roots of the teeth (Fig 2), with many covered by sediment, and some apertures on the surface of the interior cone of the tooth. The latter boring locations suggest that these teeth were isolated from the jaw at the time

of *Osedax* colonization, and the less hard tooth centres already corroded. The borings are *O. tuscia* where classifiable, and these are concentrated in one of the larger teeth (Fig 3). There are other borings in the teeth that do not closely resemble *Osspecus* and may be either exploratory worm borings or created by an entirely different organism. These are cylindrical or narrow wavy tubes often found adjacent to the *Osedax* borings and can form clusters that resemble collapsed *Osspecus* chambers (Fig 3). This specimen was dated to the Upper Cenomanian.

NHMUK PV OR 32812: a partial jaw. This specimen was previously referred to *Ichthyosaurus campylodon* and therefore likely represents *Pervushovisaurus campylodon* [31]. This specimen has previously been studied by Richard Owen [32]. The jaw is very weathered, and borings extend either into areas of thicker cortical layer or beneath it. Many of the borings are collapsed, and represent the frayed-looking hemispherical chambers of *O. eunicefootia*, with *O. tuscia* borings less commonly found (Fig 3). This specimen was dated to the Lower Cenomanian.

## Discussion

### Ichnospecies and biological species

We have erected seven new ichnospecies of *Osspecus* based on their distinctive morphologies and boring geometries, bringing the total of known *Osspecus* ichnospecies to eight. As with all trace fossils, it remains unclear to what extent individual ichnospecies represent the activities of different biological species of *Osedax* worms. Differences in boring behaviours may be in occasionally due to bone substrate type and density, i.e., teeth versus limb bone, and/or the degree of postmortem decay and bacterial colonisation. There is however strong evidence from modern examples that supports different boring morphotypes belonging to different species of *Osedax*, and there are no examples from these modern descriptions in which one species makes multiple distinct geometries [11,12]. In the fossil record, where the worm in question is never present in the boring, it is possible that convergent boring morphologies may also be overlooked, and it is possible that multiple species may use the same mode of branching to optimize nutrient exploitation. The borings may still be categorized into morphotypes with consistent geometries, however, and the evidence collected so far shows that these ichnospecies likely correlate to the number of individual biological species present in ancient vertebrate deadfalls.

Additional examples of these morphologies in the fossil record may help to refine definitions of the unique and distinct features of these ichnospecies and make them more useful for determining the diversity of ancient biological species. It is also likely that additional examples from literature previously describing either modern bone or fossil bone converge on additional borehole morphotypes that may be categorized into new ichnospecies. Further studies of the ichnotaxonomy associated with *Osedax* in the modern day will also provide additional evidence elucidating the degree to which ichnospecies and biological species have a one-to-one relationship.

### Environmental controls on preservation of *Osspecus* and implications for Cretaceous diversity

There are two central factors that are likely to determine the preservation of *Osspecus* in the fossil record: depositional environment and the environmental preferences of *Osedax*. Morphological features of the borings may also influence the preservation potential of certain morphotypes. For instance, as with *O. morsus*, where the chambers are nested relatively deep

beneath the surface of the bone, preservation potential is higher. Inversely, borings such as *O. eunicefootia*, with shallow, hemispherical chambers sitting just below the surface, are more prone to de-roofing and weathering, and less likely to be well-preserved in fossil bones.

In these marine reptile specimens, we find that there are sometimes multiple morphotypes in a single bone, and sometimes only one, with morphotypes that co-exist in one bone having varying dominance in that specimen (Supplementary Data). This pattern may reflect the dominant biological species present at that particular vertebrate fall. In today's oceans, whale-falls are at times dominated by one species [33–39] or colonized by multiple species [4,37,39,40], and these variations in colonisation may reflect fine-scale differences in environmental preference, implying that niche partitioning is occurring even within this fairly constrained ecological role. While single-species dominance is common today, up to seven species of *Osedax* have been found on the same whale skeleton [39]. Individual *Osedax* species today may prefer certain depths and oxygen levels, which provide a spatial dimension for niche availability [8,37,39]. Additional partitioning occurs on a temporal scale, with different species dominating the assemblage in succession, at differing degrees of skeletal decomposition [4,37,39,41]. If this partitioning is also reflected in the distribution of *Osspecus*, then the presence of multiple ichnospecies may indicate a longer duration on the seafloor prior to burial, and the presence of specific ichnospecies in certain bones may also indicate a finer-scale ecological preference.

Average sedimentation rates in the Chalk Sea were probably low; perhaps ~2.5 cm/ka [42]. which would imply that bone material could have spent appreciable periods exposed on the seafloor. Marine bioerosion is generally found in oxygenated environments, as the organisms that are known bioeroders of bone require some degree of oxygen to survive [43], and *Osedax* is only found in oxygenated environments today. Except possibly during deposition of the Upper Cenomanian Plenus Marls Member (corresponding to Oceanic Anoxic Event 2 [44]), the Chalk seafloor was likely fully oxygenated [45].

To date, *Osspecus* has only been found in epicontinental shelf sea environments in the Cretaceous [15,16], but this does not necessarily indicate an environmental preference, merely the prevalence of these environments in the fossil record [46]. Hay (1995) argued that the chalk is a unique low-productivity deposit representing the extension of oceanic conditions onto the continental shelf owing to breakdown of shelf-margin fronts and exceptionally high Late Cretaceous sea levels [47]. Several groups of present-day deep sea invertebrates are found in the chalk, e.g., ophiuroids [48], echinothuriid echinoids [49], and scalpellid cirripedes [50]. However, the chalk is unlikely to have been deposited in water depths greater than 200 m [51]. The Chalk Sea thus provided conditions parallel with those of the present-day deep sea, but at much shallower depths.

The additional ichnospecies described here bring the total number of ichnospecies of *Osspecus* to eight. Five of them are present in the six marine reptile specimens presented here. Previously, it was shown that three borehole morphotypes were present in one single pliosaur tooth from the UK Chalk Group [16], one of which, *O. morsus*, is found again here. Alongside an additional morphotype found in the Mons Basin of Belgium [16], new nannofossil dating of NHMUK marine reptile specimens from the UK Chalk Group indicates that *Osedax* or its osteophagous ancestors were present throughout the entire Cenomanian–Maastrichtian interval in the chalk seas of NW Europe (Fig 4).

Within the UK Chalk Group, the highest concentration of marine reptile specimens bearing *Osspecus* is in the upper Cenomanian directly before Oceanic Anoxic Event 2 (Fig 4) [32]. The highest number of co-occurring ichnospecies (five) is in this interval of UC3d (Fig 4). *O. tuscia* occurs most often throughout the Late Cretaceous, while four other ichnospecies are limited to a single marine reptile specimen (Fig 4). It should be noted, however, that previously described borings resembling *O. eunicefootia* [12], *O. tuscia* [10], *O. automedon*

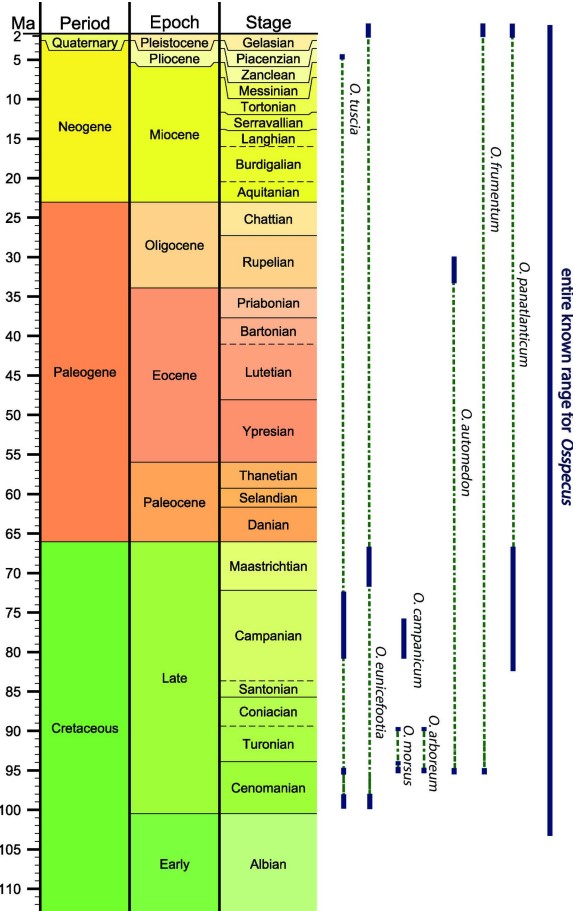

**Fig 5. The full time ranges of the eight currently described ichnospecies of *Osspecus*.** Earliest fossil occurrence from Danise & Higgs (2015) [15]. Ichnospecies occurrences are indicated by navy bars, and inferred ranges by dashed green lines. Numerical ages from GTS2020 [27] and TimescaleCreator8 (timescalecreator.org).

[14], *O. frumentum* [12], and *O. panatlanticus* [11,12] are found in the Cenozoic, (Fig 5). Both of the earliest ichnospecies found in the Lower Cenomanian in this study, *O. tuscia* and *O. eunicefootia*, are also found in the Cenozoic (Fig 5). With the available evidence, we present the time ranges for these ichnospecies as the persistence of boring behaviours producing the same morphologies through this deep-time interval rather than as an assertion of species persistence throughout this interval (Fig 5). Whether the species exhibiting these particular boring behaviours have a biological taxonomic relationship is yet to be determined.

The presence of multiple co-occurring *Osspecus* ichnospecies early in the Late Cretaceous (i.e., Cenomanian stage; 100-93.9 million years ago) likely indicates a relatively high species diversity for *Osedax* by this point in time. The occurrence of *O. eunicefootia* near the very base of the Cenomanian, with the same or very similar morphology as borings made by the living species *Osedax antarcticus*, suggests that at least extant behaviours, had appeared by 100 Ma. Based on these new data, and the oldest known *Osspecus* record from the Albian [15], it seems likely that the *Osedax* group originated in the Early Cretaceous and then diversified rapidly in the Late Cretaceous. Current molecular phylogenies for this group are not complete and contain few modern species that create described ichnotaxa [2,18]. Integrating updated molecular phylogenies with

ichnotaxonomic studies of living species and fossil specimens will strengthen divergence estimates. In turn, that will improve understanding of the relationship between fossil evidence and modern taxa.

There is currently very little available marine reptile material from the Early Cretaceous in Europe and the United Kingdom, and we suggest that Early Cretaceous material from other regions of the globe needs to be examined with our results in mind, with the added possibility of further expanding the biogeographic range of this clade in deep time.

## Conclusion and future work

We present six previously undescribed instances of *Osspecus* bioerosion in marine reptile bone derived from the Late Cretaceous Chalk Group of the UK. By characterising the broad morphological features that vary among the borings and systematically comparing them to previous descriptions, we identify eight distinct boring morphotypes. One of these corresponds to *O. tuscia*. The rest are named as seven new ichnospecies. Four of these are in the marine reptile material examined here alongside *O. tuscia,* whereas three are from material previously described elsewhere [16]. Our new dating of the marine reptiles using the nannofossil biostratigraphy of chalk matrix samples indicates that these taxa span from the Lower Cenomanian to the Upper Campanian. Boring diversity is highest in the Cenomanian and Turonian, showing that multiple species of *Osedax* were likely cohabiting in the unique pelagic setting represented by early Late Cretaceous chalk seas. This high diversity in the early Late Cretaceous is close to the supposed origin time for this clade that is based on the earliest fossil occurrence of *Osspecus* in the Albian. These results suggest either a rapid diversification or an older origination than previously thought. The persistence of certain morphotypes across the K-Pg boundary and into the present day suggests that further investigation of the borings made by extant species as well as those found in the fossil record may help to elucidate the taxonomic relationships and evolutionary history of this unique clade.

## Acknowledgements

For CT-scanning assistance we thank: Brett Clark (NHMUK), Darcy Adhami (NHMUK), Agnese Lanzetti (NHMUK), Camille Locatelli (RBINS); For curating and packing the fragile ichthyosaur jaw specimen: Lu Allington-Jones; For curating the RBINS holotype: Annelise Folie. For discussion and feedback: Phil Mannion, Paul Upchurch, Giovanni Serafini, Nicolas Bekkouche. For help with Latin and Greek nomenclature checks: Fiachra Mac Góráin. For help with Eunice Foote biographical information and etymology: Joseph Ortiz and Roland Jackson.

Lastly we would also like to acknowledge Raymond Sorenson who rediscovered Eunice Foote's work and helped promote it.

## Author contributions

**Conceptualization:** Sarah Jamison-Todd, James D. Witts.

**Data curation:** Sarah Jamison-Todd, James D. Witts, Marc E. H. Jones.

**Formal analysis:** Sarah Jamison-Todd, Deborah Tangunan, Paul Bown.

**Funding acquisition:** Paul Bown, Richard J. Twitchett.

**Investigation:** Sarah Jamison-Todd, Marc E. H. Jones.

**Methodology:** Sarah Jamison-Todd, James D. Witts, Deborah Tangunan, Kim Chandler, Paul Bown, Richard J. Twitchett.

**Project administration:** Sarah Jamison-Todd.

**Validation:** Sarah Jamison-Todd.

**Visualization:** Marc E. H. Jones.

**Writing – original draft:** Sarah Jamison-Todd.

**Writing – review & editing:** Sarah Jamison-Todd, James D. Witts, Marc E. H. Jones, Paul Bown, Richard J. Twitchett.

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
