## [Decision Letter · Decision Letter 0]

21 Jan 2025

PONE-D-24-51167A timeline for bone-eating worm diversity in the Late Cretaceous Chalk GroupPLOS ONE

Dear Dr. Jamison-Todd,

Thank you for submitting your manuscript to PLOS ONE. After careful consideration, we feel that it has merit but does not fully meet PLOS ONE’s publication criteria as it currently stands. Therefore, we invite you to submit a revised version of the manuscript that addresses the points raised during the review process.See note below. Please submit your revised manuscript by Mar 07 2025 11:59PM. If you will need more time than this to complete your revisions, please reply to this message or contact the journal office at plosone@plos.org . Please include the following items when submitting your revised manuscript:

We look forward to receiving your revised manuscript.

Kind regards,

Steffen Kiel, Ph.D.

Academic Editor

PLOS ONE

2. In your manuscript, please provide additional information regarding the specimens used in your study. Ensure that you have reported human remain specimen numbers and complete repository information, including museum name and geographic location.

'All necessary permits were obtained for the described study, which complied with all relevant regulations.

For more information on PLOS ONE's requirements for paleontology and archeology research, see https://journals.plos.org/plosone/s/submission-guidelines#loc-paleontology-and-archaeology-research.

“This study was supported in part by the NERC-funded ChaSE (Chalk Sea Ecosystems) project (NE/X015300/1, NE/X015386/1). Natural Environment and Research Council website: https://www.ukri.org/councils/nerc/. The grant was awarded to RJT and PB.”

Additional Editor Comments:

Dear Sarah,

apologies this has taken so long - one reviewer agreed to review but never did. so I'm moving forward with the one I have. If you decide to submit a revision, please address each comment in an accompanying response letter. I think you can disregard the question about Zoobank - that's for biological species only, not their traces.

Regards, Steffen

Reviewers' comments:

Reviewer's Responses to Questions

**Comments to the Author**

1. Is the manuscript technically sound, and do the data support the conclusions?

Reviewer #1: Partly

2. Has the statistical analysis been performed appropriately and rigorously? 

Reviewer #1: N/A

3. Have the authors made all data underlying the findings in their manuscript fully available?

Reviewer #1: Yes

4. Is the manuscript presented in an intelligible fashion and written in standard English?

Reviewer #1: Yes

5. Review Comments to the Author

Reviewer #1: You seem to assert that just because you cannot distinguish an ancient Osedax burrow from one made by a living species, that the same species made holes 100 mya? If that is not what you mean, then soften this part of the paper. It is one thing to say you cannot distinguish the borings, but quite another to say they were made by the same species. You cannot prove it, nor can anyone refute it, at present. Some other specific items:

Line 17...associated with whale carcasses...(what about other vertebrates?)

Ln. 24, delete 'new' and say previously undocumented

Ln. 27, these are from different depths, the WA specimens are from deep water, yours are all from shallow water?

Ln. 29-30, created by living species? Is this what you mean? Are you trying to say that living species made the Cretaceous borings? I hope not.

Ln. 36, your emphasis is on whale falls again, when Osedax is found on other bones.

Ln. 40, here you use whalefall, other places you used whale fall. I think whale-fall might be better throughout.

Ln. 64 should read 'whale, bird, and fish' and add ref. Kiel et al. 2011 (Osedax on fossil bird bones).

Ln. 74, delete the word 'instance'

Ln. 207-208, delete 'to this ichnogenus'

Ln. 239 - define outward, downward, etc. in Methods. Make sure you aren't referring to orientation of the boring with respect to ventral, dorsal, lateral sides of a bone. I think you mean with respect to the surface from which the boring starts.

Ln. 269-270, you say apertures are wide, or in others fine, etc. Why not give measurements like you do for depth, and so on. Otherwise, what exactly does 'wide' mean?

Ln. 283, should use author and date for living species (and fossil too), but not sure about journal requirements.

Ln, 325 (and others), are your species endings correct? I'm not sure. Some are ok, but others need to match the genus, I think.

Ln. 336, 'wide' but again no measurement.

Ln. 337 'chambers sit at mid-depth' - what does this mean?

Ln. 364, aperture fine - what does this mean? How small? Measure

Ln. 371, delete 'boring types than most' and just say Osspecus.

Ln. 373, found in two specimens, but I think you only listed one.

Ln. 389-390, aperture length - did you mean diameter? Not clear

Ln. 462, say 'whale bones' because whalebone can also mean baleen, which is not bone.

Ln. 593, whalefalls again.

Ln. 597, insert 'skeleton' after whale.

Ln. 670, I would transpose previous and material, so that it is 'material previously'

Ln. 795, Osedax not italicized.

Another important reference you should check is Boessenecker, R.W., and Fordyce, R.E., 2014, Trace fossil evidence of predation upon bone-eating worms on a baleen whale skeleton from the Oligocene of New Zealand. Lethaia 48: 326-337. (lists Osedax borings in many other fossils from other places).

How are the holotype borings (and others identified or discussed) noted on the bone specimens? This is not evident in your figures. How would I find them if I visited the collections? Is there a special marking - circle, square, triangle, on the bone surface?

And finally, are new ichnospecies registered on Zoobank? I am not sure, but you may want to investigate this (you didn't mention it anywhere).

6. PLOS authors have the option to publish the peer review history of their article (what does this mean? ). If published, this will include your full peer review and any attached files.

**Do you want your identity to be public for this peer review?** For information about this choice, including consent withdrawal, please see our Privacy Policy .

Reviewer #1: **Yes: ** James L. Goedert

---

## [Author Response · Author response to Decision Letter 0]

11 Feb 2025

"You seem to assert that just because you cannot distinguish an ancient Osedax burrow from one made by a living species, that the same species made holes 100 mya? If that is not what you mean, then soften this part of the paper. It is one thing to say you cannot distinguish the borings, but quite another to say they were made by the same species. You cannot prove it, nor can anyone refute it, at present."

We are in agreement that the correlation between ichnospecies and biological taxonomic relationships to species today should not be over-emphasised, as further work is needed to determine this. We have therefore tried to soften the emphasis on this connection as suggested. The relevant amendments to this section of the discussion are as follows:

Lines 576-578: we have changed the phrasing “…the evidence collected so far shows that these ichnospecies likely correlate to individual biological species in ancient vertebrate deadfalls…” to: “the evidence collected so far shows that these ichnospecies likely correlate to the number of individual biological species present in ancient vertebrate deadfalls.” This should clarify that we are not referring to specific known species but rather a broader diversity metric.

Lines 579-581: again, we try to make the distinction between ancient and modern species and emphasise the borings as a diversity metric by changing “Additional examples of these morphologies in the fossil record may help to refine definitions of the unique and distinct features of these ichnospecies” to “Additional examples of these morphologies in the fossil record may help to refine definitions of the unique and distinct features of these ichnospecies and make them more useful for determining the diversity of ancient biological species.”

Lines 584-586: After the same fashion, we have changed “…in the modern day will also provide additional evidence elucidating the ichnospecies and biological species relationship" to “…in the modern day will also provide additional evidence elucidating the degree to which ichnospecies and biological species have a one-to one relationship.

Lines 648-652: We have added a caveat to the presentation of ichnospecies ranges through time: “With the available evidence, we present the time ranges for these ichnospecies as the persistence of boring behaviours producing the same morphologies through this deep-time interval rather than as an assertion of species persistence throughout this interval (Figure 5). Whether the species exhibiting these particular boring behaviours have a biological taxonomic relationship is yet to be determined.”

Line 663: “extant behaviours, if not species” has been changed to just “extant behaviours.”

We hope that this collective adjustment of emphasis on boring behaviours as a diversity metric rather than on the deep-time persistence of individual species helps to address the reviewer’s concerns.

"Line 17...associated with whale carcasses...(what about other vertebrates?)"

This is a good point, as it is relevant to the ecology and deep-time evolution of Osedax. We have added to the abstract in lines 18-19: “The worm has also been found in a variety of other vertebrates in marine environments.”

"Ln. 24, delete 'new' and say previously undocumented"

This has been changed.

"Ln. 27, these are from different depths, the WA specimens are from deep water, yours are all from shallow water?"

This is a fair point- many of the modern and the WA Oligocene examples are from deeper-water environments, where we have no analogous rock units in the Cretaceous. We have omitted discussion of water depth because the environments of the chalk and of today’s shallow- or deep- water environments are not necessarily analogous. The water depth range of the Chalk for these examples is noted in the discussion as ~200m but more analogous to deeper sea environments than these depths are today. We have avoided direct comparisons because of the uniqueness of the Chalk Sea environment.

"Ln. 29-30, created by living species? Is this what you mean? Are you trying to say that living species made the Cretaceous borings? I hope not."

This is admittedly poor phrasing, and has been changed to “…three of which are also found in present-day whalebones.” This should remove the potential misunderstanding of a correlation to one particular living species.

"Ln. 36, your emphasis is on whale falls again, when Osedax is found on other bones."

This is an important thing to note, and we have added a clause to this sentence on lines 38-39: “…though it is also found on a variety of other bones,” and cited the Rouse 2018 paper that reviews these examples.

"Ln. 40, here you use whalefall, other places you used whale fall. I think whale-fall might be better throughout."

We now use ‘whale-fall’ throughout for the sake of consistency.

"Ln. 64 should read 'whale, bird, and fish' and add ref. Kiel et al. 2011 (Osedax on fossil bird bones)."

We do not believe the examples from the bird bones in this study to firmly fit the morphological descriptions for Osspecus. They are more ambiguous than those in the whale and fish bones, and could potentially be sponge borings.

"Ln. 74, delete the word 'instance'"

Fixed.

"Ln. 207-208, delete 'to this ichnogenus'"

Fixed.

"Ln. 239 - define outward, downward, etc. in Methods. Make sure you aren't referring to orientation of the boring with respect to ventral, dorsal, lateral sides of a bone. I think you mean with respect to the surface from which the boring starts."

The current phrasing is indeed somewhat confusing. We have tried to clarify that we do in fact mean with respect to the bone surface, and have added on lines 145-146 in Methods: “The term ‘inward’ refers to the direction penetrating into the bone, and ‘outward’ as moving away from the centre of the main boring chamber.” We have also changed the term to ‘inward’ rather than ‘downward.’

"Ln. 269-270, you say apertures are wide, or in others fine, etc. Why not give measurements like you do for depth, and so on. Otherwise, what exactly does 'wide' mean?"

These relative terms are perhaps less useful than the exact measurements we have put in Table 2. A range of apertures widths is not given due to the collapsed state of many of the borings. Holotype borings were selected based on their completeness and the presence of all the morphological features used to define the ichnospecies, and the width of the apertures for these borings relative to the other dimensions are available in Table 2. We have not indicated clearly enough where these more quantifiable dimensions are to be found, so we have added to Methods in lines 154-155, after defining our terms, the sentence: “Where proportions and features of individual ichnospecies are described in relative terms, refer to Table 2 for the exact dimensions of the holotype borings.” This is the section where other terms like ‘mid’ and ‘long’ are more quantitatively define, so hopefully this paragraph now clearly indicates in a quantitative way how to interpret the ichnospecies descriptions in a clear and reproducible way.

"Ln. 283, should use author and date for living species (and fossil too), but not sure about journal requirements."

To our knowledge this formatting is not required in this context of mentioning a species that is not having new occurrences assigned to it, but if the journal indicates this formatting is needed, we can add it in.

"Ln. 325 (and others), are your species endings correct? I'm not sure. Some are ok, but others need to match the genus, I think."

Ichnospecies names have been adjusted in accordance with common Latin usage and meaning, based on consultation with a Latin and Greek expert. This has entailed changing some of their endings throughout the manuscript, including: Osspecus eunicefootae is now Osspecus eunicefootia; Osspecus arboreus is now Osspecus arboreum; Osspecus panatlanticus is now Osspecus panatlanticum; Osspecus pollardius is now Osspecus pollardium. These changes are mainly so that the species names match the genus names in gender, especially where the word ‘specus’ is neuter in Latin, rather than falsely matching the endings.

"Ln. 336, 'wide' but again no measurement."

See previous amendments to Methods addressing similar comments.

"Ln. 337 'chambers sit at mid-depth' - what does this mean?"

The definition for ‘mid-depth’ is outlined in Methods, lines 139-140, as “‘mid’ if it sits under the surface but has an aperture neck not longer than the main chamber depth”

"Ln. 364, aperture fine - what does this mean? How small? Measure."

See previous amendments to Methods addressing similar comments.

"Ln. 371, delete 'boring types than most' and just say Osspecus."

Fixed.

"Ln. 373, found in two specimens, but I think you only listed one."

We are not sure what this refers to. There are two marine reptile specimens, 4205 and 3355, which are listed in Table 2 as having this ichnospecies.

"Ln. 389-390, aperture length - did you mean diameter? Not clear"

We meant to say aperture neck length, and have changed the phrase accordingly to relate to the chamber depth.

"Ln. 462, say 'whale bones' because whalebone can also mean baleen, which is not bone."

This has been changed for clarity.

"Ln. 593, whalefalls again".

All instances have now been changed to ‘whale-falls.’

"Ln. 597, insert 'skeleton' after whale."

Fixed.

"Ln. 670, I would transpose previous and material, so that it is 'material previously'"

Fixed.

"Ln. 795, Osedax not italicized."

This reference has been fixed.

"Another important reference you should check is Boessenecker, R.W., and Fordyce, R.E., 2014, Trace fossil evidence of predation upon bone-eating worms on a baleen whale skeleton from the Oligocene of New Zealand. Lethaia 48: 326-337. (lists Osedax borings in many other fossils from other places)."

The authors are aware of this paper, and acknowledge that it is possible that this provides examples of Osedax bioerosion. However, the morphology of the borings cannot clearly be determined from surface observation, and where posited examples of Osspecus are not described based on internal morphology through scanning, references have been omitted as non-definitive examples. This example preserves an interesting ecological interaction, but it is not clear based on the presented evidence whether the grazing was occurring on Osedax worms, substrate-colonising sponges, or other infaunal soft-bodied organisms.

"How are the holotype borings (and others identified or discussed) noted on the bone specimens? This is not evident in your figures. How would I find them if I visited the collections? Is there a special marking - circle, square, triangle, on the bone surface?"

The images of the external surfaces of the bones with arrows indicating the location of the holotype borings are on MorphoSource alongside the parent CT scans in the Supporting Information link. This is indicated in the Supporting Information section at the end of the manuscript, copied here: “Scans of the relevant specimens and images of the location in the bones of holotype borings can be found on MorphoSource at: https://www.morphosource.org/projects/000651498?locale=en”

"And finally, are new ichnospecies registered on Zoobank? I am not sure, but you may want to investigate this (you didn't mention it anywhere)."

Zoobank is a repository for biological species names, rather than trace fossil names, and therefore the ichnospecies will not be added to the repository.

Additional changes:

The title has been amended to “The evolution of bone-eating worm diversity in the Upper Cretaceous Chalk Group of the United Kingdom” to better explain the breadth and focus of the study.

An addition was put in Acknowledgements: “For help with Latin and Greek nomenclature checks: Fiachra Mac Góráin. For help with Eunice Foote biographical information and etymology: Joseph Ortiz and Roland Jackson.”

---

## [Editor Report · Decision Letter 1]

16 Feb 2025

PONE-D-24-51167R1The evolution of bone-eating worm diversity in the Late Cretaceous Chalk Group of the United KingdomPLOS ONE

Dear Dr. Jamison-Todd,

Thank you for submitting your manuscript to PLOS ONE. After careful consideration, we feel that it has merit but does not fully meet PLOS ONE’s publication criteria as it currently stands. Therefore, we invite you to submit a revised version of the manuscript that addresses the points raised during the review process. Just minor things - see below. Please submit your revised manuscript by Apr 02 2025 11:59PM. If you will need more time than this to complete your revisions, please reply to this message or contact the journal office at plosone@plos.org . Please include the following items when submitting your revised manuscript:

We look forward to receiving your revised manuscript.

Kind regards,

Steffen Kiel, Ph.D.

Academic Editor

PLOS ONE

Journal Requirements:

Additional Editor Comments:

Dear Sarah,

thank you for the revised manuscript. Overall I'm happy with your responses to the reviewer's comments, and with manuscript and image quality. I found just a few minor things that I'd like you to fix or address in some way, and then I'm happy to accept the manuscript.

Regards,

Steffen

Line 97, typo in mosaaur

436, ’nude palp #20’ has been described as Osedax ryderi by Rouse et al. 2018, p. 469.

586, should this be “one-to-one” ?

632-633, ”There is therefore now evidence for at least eight species of Osedax in the Cretaceous.” – that’s a rather strong statement considering that in the section “Ichnospecies and biological species”, you outlined various reasons why the number of Ichnospecies may not equal the number of biological species. I personally would simply remove that sentence, as this would not really change any of your subsequent conclusions.

684, for the same reason, I find this statement questionable: “These ichnotaxa represent proxies for minimum species diversity.” It could just as well be a proxy for the diversity of boring behaviors.

---

## [Author Response · Author response to Decision Letter 1]

26 Feb 2025

"Line 97, typo in mosaaur"

Fixed.

"436, ’nude palp #20’ has been described as Osedax ryderi by Rouse et al. 2018, p. 469."

This was accidentally missed, and in the manuscript nude palp 20 is now listed as Osedax ryderi accordingly.

"586, should this be “one-to-one” ?"

Fixed.

"632-633, ”There is therefore now evidence for at least eight species of Osedax in the Cretaceous.” – that’s a rather strong statement considering that in the section “Ichnospecies and biological species”, you outlined various reasons why the number of Ichnospecies may not equal the number of biological species. I personally would simply remove that sentence, as this would not really change any of your subsequent conclusions."

This criticism makes sense, particularly in light of the first reviewer’s comments. This sentence has been deleted.

"684, for the same reason, I find this statement questionable: “These ichnotaxa represent proxies for minimum species diversity.” It could just as well be a proxy for the diversity of boring behaviors."

This sentence has also been deleted.

---

## [Editor Report · Decision Letter 2]

27 Feb 2025

The evolution of bone-eating worm diversity in the Upper Cretaceous Chalk Group of the United Kingdom

PONE-D-24-51167R2

Dear Dr. Jamison-Todd,

We’re pleased to inform you that your manuscript has been judged scientifically suitable for publication and will be formally accepted for publication once it meets all outstanding technical requirements.

Kind regards,

Steffen Kiel, Ph.D.

Academic Editor

PLOS ONE
---

## [Editor Report · Acceptance letter]

PONE-D-24-51167R2

PLOS ONE

Dear Dr. Jamison-Todd,

I'm pleased to inform you that your manuscript has been deemed suitable for publication in PLOS ONE. Congratulations! Your manuscript is now being handed over to our production team.

Kind regards,

on behalf of

Dr. Steffen Kiel

Academic Editor

PLOS ONE